# A Potentially Practicable Halotolerant Yeast *Meyerozyma guilliermondii* A4 for Decolorizing and Detoxifying Azo Dyes and Its Possible Halotolerance Mechanisms

**DOI:** 10.3390/jof9080851

**Published:** 2023-08-15

**Authors:** Yue Feng, Jingru Cui, Bingwen Xu, Yifan Jiang, Chunqing Fu, Liang Tan

**Affiliations:** 1Key Laboratory of Plant Biotechnology of Liaoning Province, School of Life Science, Liaoning Normal University, Dalian 116081, China; fss990214@163.com (Y.F.); acui0315@163.com (J.C.); jiangyifan202203@163.com (Y.J.); 15560356752@163.com (C.F.); 2Dalian Center for Certification and Food and Drug Control, Dalian 116037, China; xubingwen0402@126.com

**Keywords:** *Meyerozyma guilliermondii* A4, halotolerant yeast, azo dye, halotolerance mechanisms, comparative transcriptomics approach

## Abstract

In this study, a halotolerant yeast that is capable of efficiently decolorizing and detoxifying azo dyes was isolated, identified and characterized for coping with the treatment of azo-dye-containing wastewaters. A characterization of the yeast, including the optimization of its metabolism and growth conditions, its detoxification effectiveness and the degradation pathway of the target azo dye, as well as a determination of the key activities of the enzyme, was performed. Finally, the possible halotolerance mechanisms of the yeast were proposed through a comparative transcriptome analysis. The results show that a halotolerant yeast, A4, which could decolorize various azo dyes, was isolated from a marine environment and was identified as *Meyerozyma guilliermondii*. Its optimal conditions for dye decolorization were ≥1.0 g/L of sucrose, ≥0.2 g/L of (NH_4_)_2_SO_4_, 0.06 g/L of yeast extract, pH 6.0, a temperature of 35 °C and a rotation speed of ≥160 rpm. The yeast, A4, degraded and detoxified ARB through a series of steps, relying on the key enzymes that might be involved in the degradation of azo dye and aromatic compounds. The halotolerance of the yeast, A4, was mainly related to the regulation of the cell wall components and the excessive uptake of Na^+^/K^+^ and/or compatible organic solutes into the cells under different salinity conditions. The up-regulation of genes encoding Ca^2+^-ATPase and casein kinase II as well as the enrichment of KEGG pathways associated with proteasome and ribosome might also be responsible for its halotolerance.

## 1. Introduction

Wastewaters from textile dyeing industries comprise >20% of those from all industrial fields and are usually difficult to treat due to their complex components including high concentrations of residual dyes, salts, surfactants, additives, etc. [1]. As the main organic components of textile dyeing wastewaters, dyes are classified into about 25 types based on their chemical structures [2]. Among them, azo dyes, which contain at least one azo bond (–N=N–), comprise >70% of all the commercial dyes worldwide [3]. Due to their failure to bind to fibers, about 10% of the dyestuff will be lost as effluent [4], and these dyes are harmful to aquatic life or even humans due to their high chromaticity, solubility and biotoxicity. Furthermore, byproducts derived from the degradation of azo dyes such as aromatic amines may be more toxic than the dyes themselves, and they may also be mutagenic and carcinogenic to humans [5]. Therefore, it is urgent to remove residual azo dyes from wastewaters and detoxify them before releasing them into the environment.

Many treatment technologies, which are generally classified into physical, chemical and biological processes, have been applied to remove azo dyes from wastewaters [6]; among these, biological methods are more widely used due to their advantages that are superior to physical and chemical ones, including a high treatment efficiency, a low cost and environmental friendliness [2]. As the main factor of biological processes, microorganisms are diverse, widely distributed and easy to obtain, and they are also metabolically versatile and ubiquitous [2]. Studies on the biodecolorization of azo dyes mainly focus on bacteria and fungi [2,7]. Bacteria are widely studied and used due to their extensive distribution, high metabolic activity and strong adaptability [4]. However, the toxic decolorization intermediates of azo dyes (e.g., aromatic amines) can inhibit the vast majority of bacteria, thus leading to failure in terms of purification and even an increase in biotoxicity after treatment [8]. In comparison, the enzyme-mediated degradation of azo dyes by fungi depends on non-specific and non-stereoselective enzyme systems [7]. In this case, the toxic decolorization intermediates can be further degraded and detoxified under the catalysis of ligninolytic enzymes including laccase (Lac) and two peroxidases—lignin peroxidase (LiP) and manganese peroxidase (MnP). White rot fungi, one class of filamentous fungi, are the most widely used fungi for decolorizing azo dyes and even purifying dye-containing wastewaters in bioreactors [7,9]. However, they generally grow slowly and are easily contaminated by bacteria, which restrains their further development for practical applications [10]. As a class of single-cell fungi, yeasts possess fast growth speeds and strong adaptability, like bacteria, and they also have a strong metabolic capability, like filamentous fungi [11]; thus, they have attracted more and more attention in recent years. Yeasts belonging to *Pichia* [12], *Candida* [8], *Magnusiomyces* [13], *Saccharomyces* [14], *Scheffersomyces* [15], *Cyberlindnera* [16] and *Meyerozyma* [3] have been isolated and have demonstrated the capability of decolorizing and detoxifying various azo dyes in the past 10 years. Some of them are even halotolerant species [3,12,15,16]. Though there are some yeast strains that may be potentially effective for the treatment of azo dyes, the number of corresponding yeasts is still much less than that of bacteria and filamentous fungi. Therefore, more effective yeasts should be discovered and systematically researched for further applications, especially species with more superior characteristics than the existing strains such as a higher metabolic activity, a faster growth rate, stronger adaptability to extreme environments, etc.

One important limiting factor for the biological treatment of azo dyes from textile dyeing wastewaters is the high salinity caused by the usage of high concentrations of salt (up to 100 g/L) in the dye bath to maximize the fixation of dyestuffs to fibers [17]. Many microorganisms in conventional biological processes (e.g., activated sludge systems) are inhibited or even killed under high osmotic conditions, which may result in the failure of the treatment [18]. By contrast, a group of microorganisms that can survive and keep a relatively high metabolic activity with a high salinity (which are generally defined as halophilic or halotolerant microbes) can overcome the above problem during hypersaline wastewater treatment [19]. Halophilic microorganisms (halophiles), which are usually distributed in saline or even hypersaline environments, require certain concentrations of salt to survive [20]. They are generally classified as slight, moderate and extreme halophiles according to the optimal salinity scales of 2–5%, 5–20% and 20–30% (represented by NaCl concentration, *w*/*v*), respectively [21]. There are two main strategies for halophiles to resist high permeability conditions: (1) a compatible production of organic solutes and (2) an excessive accumulation of salt (mainly Na^+^ and K^+^) in the cytoplasm to balance the salinity inside and outside of the cell [18]. On the other hand, halotolerant microorganisms can grow and metabolize under saline or hypersaline conditions; however, they do not survive depending on a certain salinity and are inhibited by higher concentrations of salt [20]. Many halotolerant and halophilic bacterial strains have been isolated from different environments and used to degrade various organic pollutants under hypersaline conditions [21,22,23,24]. Among them, the ones that can degrade azo dyes have been frequently reported since the beginning of the 21st century [23,25,26]. Meanwhile, halotolerant yeasts have also attracted more and more attention. It is reported that the morphological responses of halotolerant yeasts to a high salinity mainly include the compositional regulation of the cytoplasmic membrane and cell wall, the meristem growth, as well as pigmentation [27]. In addition, their physiological responses to high osmotic environments include the intracellular accumulation of compatible solutes and the excessive uptake of salts (mainly Na^+^ and K^+^), as well as the production of polysaccharides. Some halophilic or halotolerant yeasts have also been confirmed as effective alternatives for purifying environmental contaminants under hypersaline conditions [3,12,15,16,28,29]. However, the number of existing effective halotolerant or halophilic yeasts for pollutant treatment is still far from enough for practical application. Thus, more relevant works need to be conducted.

In the present study, a halotolerant yeast that could decolorize various acid and reactive azo dyes and tolerate relatively high salinity was isolated and identified first. A systematic characterization of the yeast was consequently performed, including the optimization of its metabolic and growth conditions, the presumption of its possible degradation pathway, the determination of the key enzymes’ activity and the detoxification effect on the target dye. Furthermore, possible halotolerant mechanisms of the yeast were proposed using a comparative transcriptomics approach. As far as it is known, this is the first systematic research on a yeast with a relatively high halotolerant ability for azo dye degradation.

## 2. Materials and Methods

### 2.1. Reagents

Seven azo dyes including four acid dyes, Acid Red B (ARB, Beijing Solarbio Science & Technology Co., Ltd., Beijing, China), Acid Orange II (AOII, J&K Scientific Ltd., Shanghai, China), Acid Scarlet GR (GR, Sigma-Aldrich Company Ltd., Beijing, China) and Acid Red 3R (3R, Sigma-Aldrich Company Ltd., Beijing, China), and three reactive dyes, Reactive Brilliant Red K-2G (K-2G, Sigma-Aldrich Company Ltd., Beijing, China), Reactive Violet KN-4R (KN-4R, Dye Synthesize Laboratory of Dalian University of Technology, Dalian, China) and Reactive Yellow 3R (3R, Dye Synthesize Laboratory of Dalian University of Technology, Dalian, China), were used in this study. These seven dyes were selected in this study due to their high water solubility. The purity of these azo dyes was >98.0%. Chemical information of these dyes including chemical structure, molecular weight, characteristic absorption wavelength and CAS number are shown in the Appendix A. Other chemical and biological reagents were purchased from Tianjin Kemiou Chemical Reagent Co., Ltd. (Tianjin, China) and Beijing Solarbio Science & Technology Co., Ltd. (Beijing, China), respectively.

### 2.2. Isolation and Identification of Halotolerant Azo-Dye-Degrading Yeast

The pure culture was isolated from the sea mud sample collected from Dalian, China (38.88° N, 121.57° E) using the spread plate method and was identified through the 26S rDNA/ITS sequencing method, as described by Li et al. [3]. The selective liquid culturing medium contains the following (g/L): glucose 2.0, yeast extract 0.1, (NH_4_)_2_SO_4_ 1.0, K_2_HPO_4_ 1.0, MgSO_4_ 7H_2_O 0.5, NaCl 30.0 and azo dye 0.02. The solid medium contains an additional 2% (*w*/*v*) agar powder on the basis of the above liquid medium components. The culturing conditions were as follows: rotation speed of 160 rpm, temperature of 35 °C and initial pH of 6.0. Detailed description is shown in Appendix A.

### 2.3. Conditional Optimization of the Target Yeast for Dye Decolorization and Its Cell Growth

Firstly, the decolorization efficiency of different azo dyes with the same substance concentration (40 μmol/L) was compared by growing cells of the target yeast, and the dye, which was decolorized with the highest rate, was selected as the target dye for further investigation. Conditions for dye decolorization performance and growth of the yeast were optimized through batch tests in 250 mL shaking flasks (the volume of the actual reaction system was 100 mL). The optimized conditions included the type (sucrose, glucose, fructose, maltose, lactose, soluble starch and sodium acetate) and concentration (0–3.0 g/L) of additive carbon source, the concentration of additive nitrogen source ((NH_4_)_2_SO_4_, 0–0.6 g/L) and the vitamin mixture source (yeast extract, 0–0.1 g/L), salinity (NaCl concentration, 0–100.0 g/L), initial pH (3.0–10.0), rotation speed (0–200 rpm), temperature (25–45 °C) and concentration of the target dye (100.0–1600.0 μmol/L). Except for the investigated parameter, others were set as sucrose 2.0 g/L, (NH_4_)_2_SO_4_ 0.4 g/L, yeast extract 0.06 g/L, NaCl 30 g/L, target dye 100.0 μmol/L, rotation speed of 160 rpm, temperature of 35 °C and initial pH of 6.0, which were selected through pre-experiments. The density of the cell suspension for inoculation, which was analyzed via spectrophotometry and presented by the absorbance at 600 nm (OD_600_), was about 0.325 ± 0.005 (after 10-fold dilution). Concentrations of the azo dyes and yeast cell suspension were analyzed spectrophotometrically according to the method described by Li et al. [3].

### 2.4. Acute Toxicity Assessment

Acute toxicity of the target dye and its degradation byproducts were analyzed using the Microtox method [30]. The result is represented by the luminescence inhibition ratio (IR) of the luminescent bacteria *Vibrio fischeri* (NRRL B-11177). IRs in the ranges of 70–100%, 50–70%, 20–50%, 10–20% and 0–10% represent high, moderate, low, micro and non-toxic levels, respectively.

### 2.5. Activity Determination of Key Enzymes

Activity of five enzymes that were probably responsible for the decolorization of azo dyes and further degradation of their metabolic intermediates, including azo reductase (AZR), nicotinamide adenine dinucleotide-dependent 2,6-dichlorophenol indophenol (NADH-DCIP) reductase, lignin peroxidase (LiP), manganese peroxidase (MnP) and laccase (Lac), was analyzed via spectrophotometry. Detailed descriptions of analysis protocols and conditions are shown in a previous report [12].

### 2.6. Presumption of Degradation Pathway of the Target Dye

A possible degradation pathway of the target dye by the yeast was presumed based on the analysis of degradation intermediates and the relevant literature. The change in the main functional groups of the target dye was first analyzed using the UV-Vis scanning function of spectrophotometry (JASCO V-560, Tokyo, Japan). Then, possible degradation intermediates were determined using high-performance liquid chromatography—mass spectrometry (HPLC-MS) methods with an Agilent 1260 Infinity Bio-inert Quaternary LC System coupled to an Agilent 6130B Single Quadrupole LC/MS System (Agilent Technologies Inc., Santa Clara, CA, USA). Detailed method descriptions and experimental conditions were the same as those of Tan et al. [15].

### 2.7. Transcriptomic Responses of the Yeast to Salt Stress

A comparative transcriptomics approach was used to reveal possible halotolerant mechanisms of the yeast. Three groups (each group was set up in triplicate) of the yeast were cultured for 12 h with 0.0 g/L, 30.0 g/L and 80.0 g/L NaCl, which were named as S0, S1 and S2, respectively. The other culturing conditions were sucrose 2.0 g/L, (NH_4_)_2_SO_4_ 0.4 g/L, yeast extract 0.06 g/L, target dye 100.0 μmol/L, rotation speed of 160 rpm, temperature of 35 °C and initial pH of 6.0. After the 12 h cultivation, the yeast cells were gathered through centrifugation at 10,000× *g* and 4.0 °C for 10 min. Then, the yeast cells were washed with 0.02 mol/L phosphate buffer (pH = 7.2) for three times and subsequently cryopreserved in liquid nitrogen. The transcriptome sequencing was performed by Novogene Bioinformatics Technology Co., Ltd. (Beijing, China). Details of experimental protocols and conditions including total RNA extraction, cDNA library preparation, sequencing and bioinformatics analysis of the sequencing results are described in Appendix A. Focus was placed on the differentially expressed genes (DEGs) between the groups “S1 (the experimental group) vs. S0 (the control group)” and “S2 (the experimental group) vs. S0 (the control group)”, especially the ones related to halotolerance. The fold change of the DEGs was represented as the log_2_ fold change (log_2_ FC) of gene abundance via comparison of the experimental group and the control, with the screening criterion of 1.0. The reliability of important relevant DEGs was validated using the quantitative real-time polymerase chain reaction (QRT-PCR) method.

### 2.8. Statistical Analysis

Statistical analysis of experimental data was performed through the one-way analysis of variance (ANOVA) method using Microsoft Excel 2019. A *p*-value of less than 0.05 indicated a significant difference between the experimental group and the control (or another group).

## 3. Results and Discussion

### 3.1. Isolation and Identification of the Halotolerant Yeast Strain Capable of Decolorizing Various Azo Dyes

One yeast strain, which was named A4 and capable of decolorizing different azo dyes under high salt conditions (containing 30.0 g/L NaCl), was isolated from a marine environment (sea mud sample). Its cells were rod-shaped (Appendix A), and its colonies on the solid culture medium were circular shaped with a regular edge, rough on the surface, light yellow on the edge and red in the center area (Appendix A).

Its partial 26S rDNA sequence showed 100% homology to *Meyerozyma guilliermondii* PYCC 4738 (MZ390155), and its Internal Transcribed Spacer (ITS) sequence showed 100% homology to *M. guilliermondii* PZ03 (MK224832). Thus, the yeast A4 was identified as *M. guilliermondii*. Its partial 26S rDNA (525 bp) and ITS (577 bp) sequences were deposited to the GenBank database (https://www.ncbi.nlm.nih.gov/genbank/ accessed on 30 March 2022) with the accession numbers of ON102041 and ON089018, respectively. Based on the released 26S rDNA sequences of some yeast strains in the GenBank database and those of the yeasts we previously isolated and reported, a phylogenetic tree was constructed (Appendix A).

Growing cells of the yeast *M. guilliermondii* A4 could efficiently decolorize seven different azo dyes (40.0 μmol/L) with decolorization percentages of 86.79–99.54% within 12 h (Figure 1(a)). Among the seven azo dyes, ARB, 3R, AOII, GR and K-2G were probably decolorized mainly through biodegradation; by contrast, KN-4R and 3RS were probably decolorized through both biodegradation and biosorption, according to the colors of the colonies after culturing (Figure 1(d)). Among the seven dyes, ARB was decolorized at the highest rate, followed by 3R, AOII, KN-4R, GR, K-2G and 3RS. Thus, ARB was selected as the target dye for further study. Meanwhile, the seven tested dyes displayed little effect on the cell growth of the yeast A4 (Figure 1B).

### 3.2. Optimization of Conditions for Dye Decolorization by the Yeast A4 and Its Cell Growth

Nutrients such as carbon and nitrogen sources were required for microorganisms to efficiently degrade organics, especially the recalcitrant ones like azo dyes [31]. As shown in Figure 2A and Appendix A, the yeast A4 decolorized ARB and grew fast with sucrose, glucose or fructose; by contrast, the dye decolorization efficiency and the cell growth rate were much lower with maltose, lactose, soluble starch or sodium acetate. Among the seven tested additive carbon sources, sucrose induced the highest rate of dye decolorization and cell growth, and thus was selected for further investigation. The effect of the concentration of the additive carbon, nitrogen and vitamin mixture (provided by the yeast extract) sources on the yeast A4 was subsequently investigated. It is suggested in Figure 2B–D that more than 97.0% of 100 μmol/L ARB was decolorized by the yeast A4 with at least 1.0 g/L of sucrose, 0.2 g/L of (NH_4_)_2_SO_4_ and 0.06 g/L of yeast extract. Meanwhile, the higher concentration of sucrose resulted in faster cell growth (Appendix A); however, the higher concentration of (NH_4_)_2_SO_4_ (0.2–0.6 g/L) and yeast extract (0.06–0.10 g/L) did not obviously improve the growth of the yeast A4 (Appendix A). The effect of salinity on *M. guilliermondii* A4 is shown in Figure 2E and Appendix A. More than 97.0% of 100 μmol/L ARB was decolorized within 12 h with 0–60.0 g/L of NaCl. Even when the NaCl concentration increased to 70.0–80.0 g/L, the decolorization percentages could also achieve higher than 95.0% within 14 h, which suggested that the yeast A4 could keep relatively high metabolic activity under high salt conditions. Meanwhile, the higher concentration of NaCl displayed an inhibitory effect on both the dye decolorization and cell growth, which suggested that the *M. guilliermondii* A4 was a halotolerant yeast rather than a halophilic one. In addition, the highest rate of dye decolorization and cell growth was achieved with a pH value of 6.0, a temperature of 35 °C and a rotation speed of ≥160 rpm (Figure 2F–H and Appendix A), which suggested that *M. guilliermondii* A4 was an aerobic, mesophilic and neutrophilic yeast. At last, the effect of the dye concentration on the yeast A4 is shown in Figure 2I and Appendix A. Decolorization percentages higher than 97.0% were achieved within 16 h for 100.0–800.0 μmol/L ARB. When the initial ARB concentration further increased to 1000.0–1600.0 μmol/L, the 16 h decolorization percentages could also achieve >66.0%. Meanwhile, the average decolorization rate within 12 h increased from 8.26 μmol/(L h) to 78.54 μmol/(L h), with the initial ARB concentration increasing from 100.0 μmol/L to 1400.0 μmol/L (Appendix A). When the initial ARB concentration further increased to 1600.0 μmol/L, the 12 h average decolorization rate started to decrease to 77.85 μmol/(L h), which suggested that an ARB concentration higher than 1400.0 μmol/L displayed an inhibitory effect on the metabolic activity of the yeast A4. On the other hand, a higher ARB concentration resulted in a slower cell growth rate due to the increasing toxicity from ARB and its metabolic intermediates.

In the past ten years, we have isolated and reported nine yeast strains that are capable of decolorizing and detoxifying azo dyes, seven of which are halotolerant types [3,5,8,12,13,15,16,32,33]. At least 2.0 g/L of glucose or sucrose was required for the efficient dye decolorization for these yeasts, except for the one (*M. guilliermondii* A3) reported in 2022, which required at least 1.0 g/L glucose or sucrose [3]. The yeast A4 could also efficiently decolorize azo dyes with only 1.0 g/L glucose or sucrose, which was important for saving the addition of nutrients for practical application. In addition, all of the nine yeasts could keep a relatively high dye decolorization efficiency with ≤50.0 g/L of NaCl; by contrast, the yeast A4 could tolerate up to 80.0 g/L of NaCl. It was suggested that the yeast A4 possessed a higher halotolerant ability than those reported previously. As reported in [34], the salinity of dye-containing wastewater could be as high as 150.0–200.0 g/L of total soluble solid (TSS). Therefore, it was significantly meaningful to exploit effective microbes with a higher halotolerant ability for treating hypersaline wastewaters.

### 3.3. The Detoxification Effect of the Yeast A4 on ARB and Its Metabolic Intermediates

As shown in Figure 3, the acute toxicity of 200.0 μmol/L ARB belonged to the high toxic level based on its IR of 74.5 ± 2.87%. After being treated by the yeast A4 for 12 h, the IR increased to 85.33 ± 2.67%, suggesting that the acute toxicity increased compared with the original dye. However, the IR obviously decreased to 28.56 ± 0.95% (low toxic level) after being treated for another 12 h. This result suggests that some metabolic intermediates with a higher acute toxicity than ARB were generated and accumulated during the first 12 h, and then were decomposed to lowly toxic products during the following 12 h. The conclusion was consistent with those in previous reports [5,33]; however, it was contrary to some of the other reports that stated that the acute toxicity decreased all along the whole dye degradation process [3,16]. The possible reason might be the competition between the generation and further degradation of the toxic dye decolorization intermediates in different yeasts. Concretely, the decrease in acute toxicity within 12 h might be attributed to the higher degradation rate of aromatic amines than that of the corresponding dyes by some yeasts. By contrast, if the degradation rate of the azo dyes was higher than that of the corresponding aromatic amines, the acute toxicity would increase within 12 h due to the accumulation of aromatic amines. These aromatic amines would be further degraded during the following 12 h, thus resulting in a decrease in the acute toxicity.

### 3.4. Determination of Key Enzymes’ Activity

The activities of the five key enzymes under different conditions were only determined intracellularly, and the results are shown in Figure 4. It is shown that the activities of all five key enzymes were higher in the presence of ARB than the corresponding ones without ARB. Furthermore, ARB significantly raised the activities of the NADH-DCIP reductase, LiP, MnP and Lac according to the corresponding *p*-values, which is consistent with the conclusions of previous reports [3,35]. On the other hand, 30.0 g/L of NaCl significantly inhibited the activities of AZR and MnP compared with those without NaCl, which is also consistent with the results of a previous study [3]. However, 30.0 g/L of NaCl induced the increase in the activities of the NADH-DCIP reductase (significantly), LiP (non-significantly) and Lac (non-significantly), which might be related to the salt-induced increase in the expression level of the corresponding enzymes [36]. In addition, when the NaCl concentration further increased to 80.0 g/L, the activities of all five key enzymes were lower (significant: AZR, NADH-DCIP reductase, MnP and Lac; non-significant: LiP) than the corresponding ones without NaCl, suggesting that a higher salinity displayed an inhibitory effect on all the key enzymes.

### 3.5. Degradation Pathways Presumption of ARB by the Yeast A4

Appendix A shows the UV-Vis scanning diagram of ARB (200.0 μmol/L) before and after being treated by the yeast A4 for 12 h and 24 h. The absorbances at 516 nm and 322 nm corresponded to ARB’s azo bond and naphthalene ring structure, respectively, and decreased after biodegradation. However, the absorbance at 238 nm (which represents the benzene ring structure) increased during the first 12 h, then decreased during the following 12 h. It was suggested that ARB was decomposed into aromatic intermediates, which were initially accumulated and then further degraded by the yeast A4. In order to presume a more detailed degradation pathway, six possible intermediates generated from ARB degradation were determined through HPLC-MS analysis (Appendix A), and the possible degradation pathway (Figure 5) was proposed based on the HPLC-MS results and the relevant literature. The decolorization of ARB by the yeast A4 probably started at the cleavage of the azo bond to generate the corresponding aromatic amines, which are generally catalyzed by AZR, NADH-DCIP reductase and/or Lac [37]. The detection of compound II could verify this presumption. Compound II might be transformed into compound IV through the oxidative hydroxylation pathway, and then compounds I and IV might be further transformed into compound III (detected) through the oxidative deamination pathway. Subsequently, compound III might be transformed into compound VI (detected) through the reductive dehydrogenation pathway. The above steps were the same as those in the previous report in 2022 [3]. Meanwhile, compound III might also be transformed into compound V (detected) through the oxidative desulfurization pathway, and then compound V might be transformed into compound VII, which was also the possible secondary metabolite of compound VI. The pathway from compounds III through V to IV was the same as that in another study [32]. Compound VII might be transformed into compound VIII through a series of steps [38]. Furthermore, compound VIII might be transformed into compounds IX and X through the open loop of the benzene ring and then might be finally mineralized through the TCA cycle. Peroxidases such as LiP and MnP, and oxidoreductases such as Lac might play important roles in the oxidative transformation of aromatic amines and their downstream byproducts [5]. Compared with our previous research, the downstream section (from compound VII to the end) of the whole presumed process displayed many more details due to the detection of compounds IX and X. However, this presumed that the degradation pathway might still be imprecise in some details, and thus needs to be further discussed with the help of more advanced analytical methods.

### 3.6. Possible Halotolerant Mechanism of the Yeast A4 Revealed via Comparative Transcriptome Analysis

#### 3.6.1. Transcriptome Sequencing Results

After the quality control of the raw RNA sequences, 144,409,984 (21.65 Gb), 148,506,792 (22.27 Gb) and 148,871,154 (22.34 Gb) clean reads corresponding to the samples S0, S1 and S2 were obtained, with the average GC contents of 45.43%, 45.47% and 45.46, respectively. The average Q20/Q30 percentages were 97.90%/93.87%, 98.00%/94.03% and 97.85%/93.66%, and the average matching percentages of clean sequences to the reference genome were 98.48%, 98.59% and 98.34% corresponding to the samples S0, S1 and S2, respectively.

#### 3.6.2. DEGs Associated with Halotolerance and Other Related Responses

The DEGs between the samples S1/S2 (the experimental group) and S0 (the control) were analyzed and discussed for revealing possible halotolerant mechanisms of the yeast A4. There were 609/1247 up-regulated genes, and 347/841 down-regulated genes were determined as DEGs between the samples S1/S2 and S0 (Appendix A).

The DEGs related to the halotolerance of the microorganisms were first focused on. As reported, the regulation of the cell wall components was one of the responses for many yeasts to resist high osmotic conditions [3]. In this study, seven genes encoding the 1,3-beta-glucan synthase component (D_105169 and D_105067), endo-1,3(4)-beta-glucanase (D_104046), the cell wall protein ECM33 (D_104112), the cell surface GPI-anchored protein ECM33 (D_101126) and chitin synthases (D_101744 and D_100586) were determined to be up-regulated (Table 1) due to 30.0 g/L of NaCl (in the “S1 vs. S0” comparison). Meanwhile, no down-regulated DEG-encoding proteins or other components of the cell wall were detected. When the NaCl concentration increased to 80.0 g/L, six up-regulated DEGs (D_102776, D_105169, D_101126, D_101744, D_100803 and D_103766) encoding possible cell wall components were determined in S2 compared to S0 (in the “S2 vs. S0” comparison), and still no relevant down-regulated DEG was detected. These up-regulated DEGs might be related to the enhancement of cellular elasticity and rigidity, thus protecting the yeast cells from external high osmotic pressure [39,40,41,42]. It was suggested that the cell wall component regulation was also one important strategy to resist high salinities for the yeast A4, which was consistent with the suggestions of previous studies [3,5,33,43].

As reported by Oren [44], halophilic or halotolerant microorganisms could tolerate high osmotic pressure by balancing the intracellular and extracellular salinities through the excessive uptake and accumulation of K^+^ and/or Na^+^ in their cells, which was defined as the “high-salt-in strategy”. In this study, two genes encoding high-affinity potassium transporter (D_102974) and low-affinity K^+^ transporter 1 (D_104268) were up-regulated in S1 and S2 compared to S0, respectively. It was suggested that the “high-salt-in strategy” was activated by both 30.0 g/L and 80.0 g/L of NaCl [45].

On the other hand, the halophiles and halotolerant microbes could also resist high osmotic conditions through the production and intracellular accumulation of compatible solutes such as polyols, sugars, amino acids, etc. [46]. This mechanism was defined as the “organic-solutes-in strategy” [44]. Glycerol is usually utilized as an osmolyte by many yeasts. It was confirmed that higher glycerol-3-phosphate dehydrogenase activity resulted in more glycerol production in the halotolerant yeast *Pichia farinose* [47], suggesting that the up-regulation of genes encoding the glycerol-3-phosphate dehydrogenase might be responsible for the enhancement of halotolerance. Two genes encoding the glycerol-3-phosphate dehydrogenase were determined as DEGs. One (D_102964) was down-regulated in S1, and the other one (D_102983) was up-regulated in S2, compared with S0. Similarly, another gene (D_103937) encoding NADP-specific glutamate dehydrogenase, which was responsible for synthesizing a compatible solute for resisting high osmotic conditions [48], was also only determined to be up-regulated in the “S2 vs. S0” comparison. It was suggested that the “organic-solutes-in strategy” was probably only activated by 80.0 g/L of NaCl.

In addition, other halotolerance mechanisms related to Ca^2+^-ATPase and casein kinase II were also reported [49,50]. Park et al. [51] found that the *PMR1* gene encoding a *P*-type Ca^2+^-ATPase in one mutation of the yeast *Saccharomyces cerevisiae* was responsible for the enhancement of halotolerance through continuously activating calcineurin and enhancing the expression of the *PMR2* gene encoding a *P*-type Na^+^-ATPase. de Nadal et al. [49] also showed that the expression of Ca^2+^-ATPase in *Saccharomyces cerevisiae* enhanced its halotolerant ability. On the other hand, casein kinase II was proposed as an important additional component of the Trk1-Trk2 transport system, which could maintain the halotolerance of *Saccharomyces cerevisiae* by inhibiting excessive Na^+^ entry into cells [50]. Three up-regulated DEGs encoding calcium-transporting ATPase (D_101005 and D_103758) and casein kinase II subunit alpha (D_101154) were determined in the “S2 vs. S0” comparison; however, none of them were detected in the “S1 vs. S0” comparison. It was suggested that the corresponding halotolerant strategy might only be activated by a higher salinity (80.0 g/L NaCl).

As shown in Figure 6, the activities of AZR and MnP were inhibited by both 30.0 g/L and 80.0 g/L of NaCl. Meanwhile, though the activities of the NADH-DCIP reductase, LiP and Lac were inhibited by 80.0 g/L of NaCl, they were enhanced by 30.0 g/L of NaCl. In the transcriptome sequencing results, two genes (D_103507 and D_103508) encoding probable NADPH dehydrogenase (NADH:flavin oxidoreductase) were determined to be down-regulated in the “S1 vs. S0” comparison, and one of them (D_103507) was also determined to be down-regulated in the “S2 vs. S0” comparison. AZR belonged to the NADH-dependent flavin oxidoreductase family [52]. Thus, the down-regulation of genes encoding probable NADPH dehydrogenase (NADH:flavin oxidoreductase) was consistent with the decrease in AZR activity with a higher salinity. One gene (D_103084) encoding yeast manganese transporter smf1 was down-regulated in both S1 and S2 compared with S0. As reported, the increase in MnP activity was attributed to the overexpression of the manganese transporter smf2 homologue gene *PsMnt* in *Phanerochaete sordida* YK-624 [53], suggesting that the down-regulation of D_103084 might also be consistent with the decrease in MnP activity in the presence of NaCl. In addition, another gene (D_100122) encoding laccase PFICI_06862 was up-regulated in the “S1 vs. S0” comparison; however, it was down-regulated in the “S2 vs. S0” comparison. This result was also consistent with the increase and decrease in Lac activity in the presence of 30.0 g/L (S1) and 80.0 g/L (S2) of NaCl, respectively. Though no DEG encoding LiP was found in any of the three groups of samples, one gene (D_104858) encoding cytochrome c peroxidase, which possessed a similar three-dimensional structure and function with LiP [54], was determined to be up-regulated in the “S1 vs. S0” comparison and down-regulated in the “S2 vs. S0” comparison. Unfortunately, no gene encoding NADH-DCIP reductase was determined in any of the samples. As described by Wang et al. [55], the current genome annotation of some lower organisms was still incomplete because their genomes might have been established early and had not been updated for a long time. Thus, the possible reason might be the incomplete annotation of the reference genome.

As shown in Figure 2E and Appendix A, both the dye decolorization performance and the cell growth of the yeast A4 were inhibited by the higher concentration of NaCl; thus, relevant DEGs were also concerned. Hexoses such as sucrose, glucose or fructose were required for both the metabolism and growth of the yeast A4 (Figure 2A and Appendix A). It was indicated that the transportation and utilization of sugar were important for co-metabolism processes and microbial growth [5,56]. In this study, some genes encoding sugar transporters were determined (Appendix A). Among them, only one gene (D_104547) encoding hexose transporter 2 was up-regulated in both of the two comparisons. In addition, another two (D_102607 and D_102333) genes and one (D_104584) gene encoding sugar transporters were also determined to be up-regulated in the “S1 vs. S0” and “S2 vs. S0” comparisons, respectively. By contrast, five and sixteen homogeneous genes were detected to be down-regulated in the “S1 vs. S0” and “S2 vs. S0” comparisons, respectively, which were more than the corresponding up-regulated ones. This might be one reason for the decrease in metabolism and growth activity of the yeast A4 under high salt conditions. On the other hand, cytochrome P450 was confirmed as a reductase that is involved in azo dye biodegradation [57]. Only one gene (D_101278) encoding cytochrome P450 was up-regulated in both of the two comparisons. Other DEGs encoding cytochrome P450 were all down-regulated in S1 and/or S2 compared with S0, suggesting that the azo dye reduction activity was inhibited by high salinities. However, another gene (D_104882) encoding NADH-ubiquinone oxidoreductase assembly factor N7BML was up-regulated in the “S2 vs. S0” comparison, which might be one of the reasons why the dye decolorization efficiency did not decrease much in the presence of 80.0 g/L of NaCl.

#### 3.6.3. KEGG Enrichment

The significantly enriched KEGG pathways were primarily associated with proteasome, ribosome biogenesis in eukaryotes, DNA replication, ribosome and mismatch repair in the “S1 vs. S0” comparison, and ribosome biogenesis in eukaryotes, proteasome, RNA polymerase, spliceosome and peroxisome in the “S2 vs. S0” comparison (Figure 7). The pathway associated with ribosome biogenesis in eukaryotes was significantly enriched in both of the two comparisons, suggesting that the activity of ribosomes in the yeast cells might be greatly enhanced to meet the rapid synthesis of proteins for adapting to the high osmotic environment as soon as possible [58]. Meanwhile, the enrichment of pathways associated with DNA replication (17 up-regulated and 0 down-regulated genes in the “S1 vs. S0” comparison) and RNA polymerase (21 up-regulated and 0 down-regulated genes in the “S2 vs. S0” comparison) might also be responsible for the adaptation to high salinities. In addition, the ubiquitin–proteasome pathway played an important role in controlling the various cellular processes of yeasts, including the responses to salt stress [59]. Thus, the significant enrichment of the KEGG pathway associated with proteasome in both of the two comparisons suggested that it might be another important response of the yeast A4 for adapting to high salt environments. Furthermore, peroxisomes were also confirmed as crucial for resisting salt stress in yeasts [60]. The pathway associated with peroxisome was significantly enriched only in the “S2 vs. S0” comparison, suggesting that peroxisomal activity might be important for the yeast to tolerate a higher salinity (80.0 g/L NaCl).

## 4. Conclusions

A halotolerant yeast that could decolorize various azo dyes and tolerate up to 80.0 g/L of NaCl was isolated and identified as *Meyerozyma guilliermondii* A4. The strain A4 is an aerobic, mesophilic and neutrophilic yeast, and its growing cells could decolorize more than 97.0% of 100 μmol/L ARB under the following conditions: ≥1.0 g/L of sucrose, ≥0.2 g/L of (NH_4_)_2_SO_4_, 0.06 g/L of yeast extract, a pH value of 6.0, a temperature of 35 °C and a rotation speed of ≥ 160 rpm. The yeast A4 decolorized and detoxified the ARB through a series of steps including the cleavage of the azo bond, oxidative hydroxylation, oxidative deamination/desulfurization, reductive dehydrogenation, the open loop of the benzene ring, and the TCA cycle, relying on the key enzymes such as AZR, NADH-DCIP reductase, LiP, MnP and Lac. The transcriptomic responses of *M. guilliermondii* A4 to different salinities suggested that the yeast tolerated 30.0 g/L and 80.0 g/L of NaCl through the regulation of the cell wall component and the accumulation of K^+^/Na^+^ in cells; meanwhile, other strategies such as the production and intracellular accumulation of compatible solute and the overexpression of Ca^2+^-ATPase and casein kinase II were only activated in the presence of 80.0 g/L of NaCl. The KEGG pathways associated with proteasome, ribosome biogenesis in eukaryotes, DNA replication, RNA polymerase and peroxisome were also enriched for resisting high osmotic conditions.

## Figures and Tables

**Figure 1 jof-09-00851-f001:**
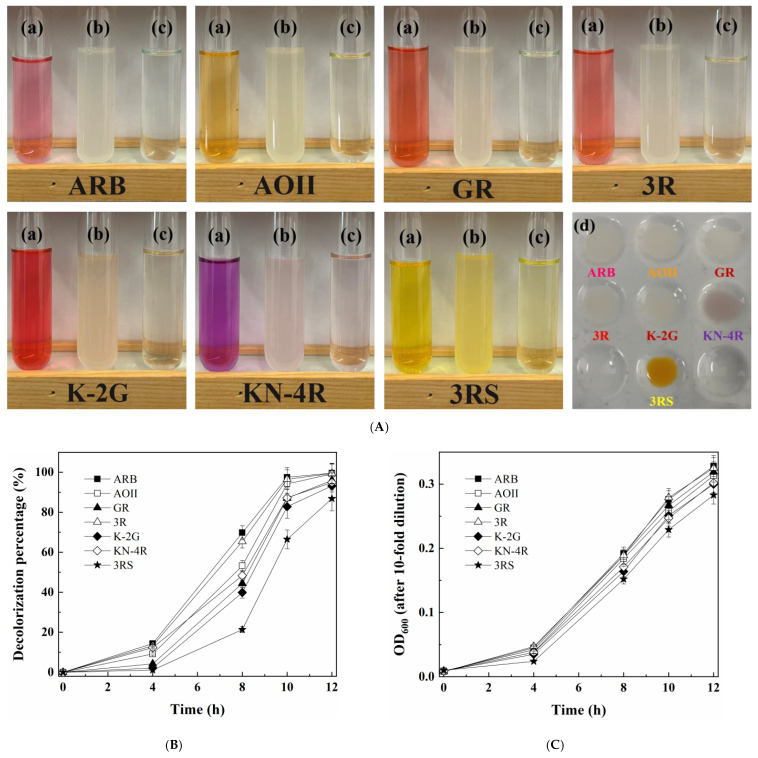
(**A**) Photos of (a) original dye solution of seven azo dyes (40 μmol/L); (b) cell suspension, (c) supernatants and (d) yeast cells after decolorization by growing cells of *M. guilliermondii* A4. (**B**) Decolorization and (**C**) cell growth curves during dye decolorization by *M. guilliermondii* A4.

**Figure 2 jof-09-00851-f002:**
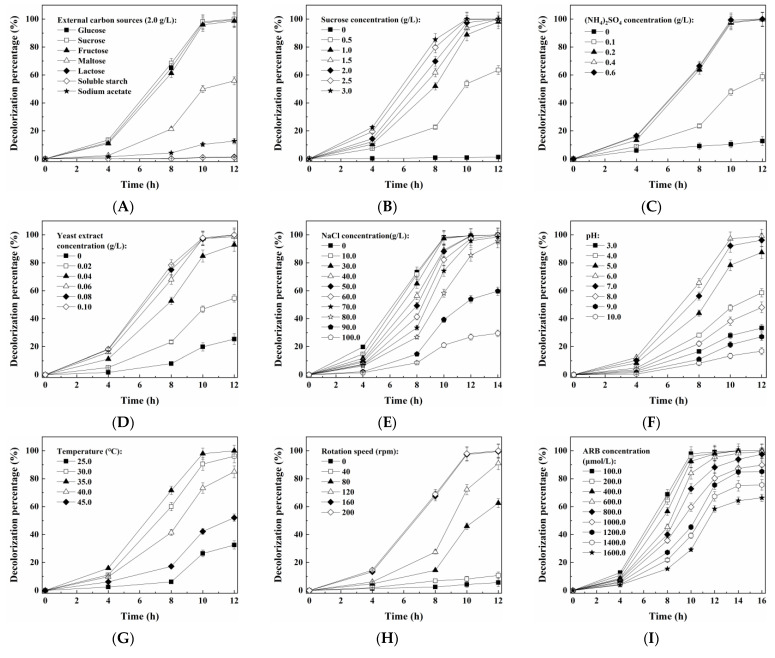
Optimization of the conditions for ARB decolorization by growing cells of *M. guilliermondii* A4: (**A**) type of external carbon source (2.0 g/L); (**B**) sucrose concentration; (**C**) (NH_4_)_2_SO_4_ concentration; (**D**) yeast extract concentration; (**E**) NaCl concentration; (**F**) pH; (**G**) temperature; (**H**) rotation speed and (**I**) ARB concentration.

**Figure 3 jof-09-00851-f003:**
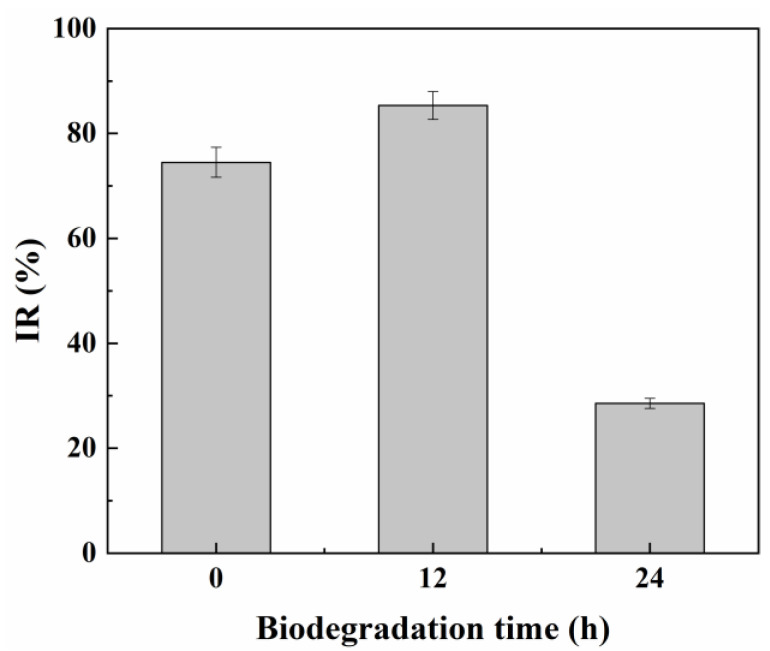
Acute toxicity (using Microtox method) of 200.0 μmol/L ARB before and after being treated for different times (12 h and 24 h) by cell growth of *M. guilliermondii* A4. A value of 70 ≤ IR (%) < 100 represents high toxicity; 50 ≤ IR (%) < 70 represents moderate toxicity; 20 ≤ IR (%) < 50 represents low toxicity; 10 ≤ IR (%) < 20 represents micro toxicity and 0 ≤ IRs (%) < 10 represents non-toxicity.

**Figure 4 jof-09-00851-f004:**
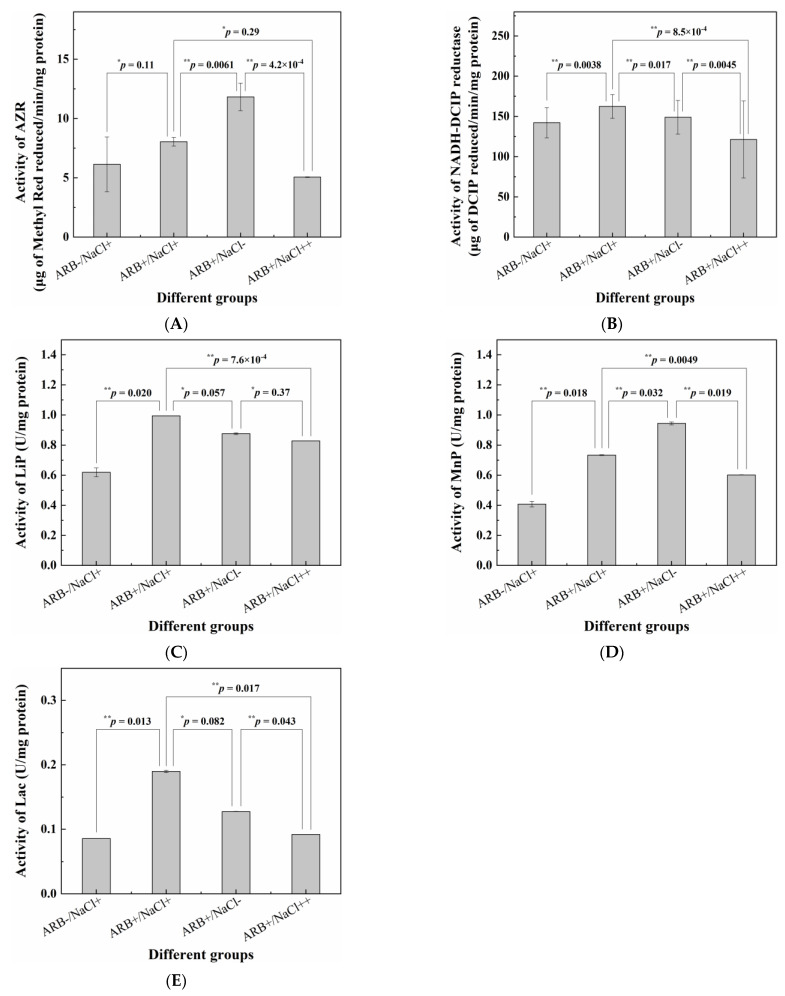
Activity of five key enzymes under different conditions: (**A**) AZR; (**B**) NADH-DCIP reductase; (**C**) LiP; (**D**) MnP and (**E**) Lac. “ARB−” and “ARB+” represent the culturing conditions without and with 100.0 μmol/L ARB, respectively; “NaCl−”, “NaCl+” and “NaCl++” represent the culturing conditions with 0 g/L, 30.0 g/L and 80.0 g/L NaCl, respectively. * and ** represent the corresponding *p*-values of >0.05 and ≤0.05, respectively.

**Figure 5 jof-09-00851-f005:**
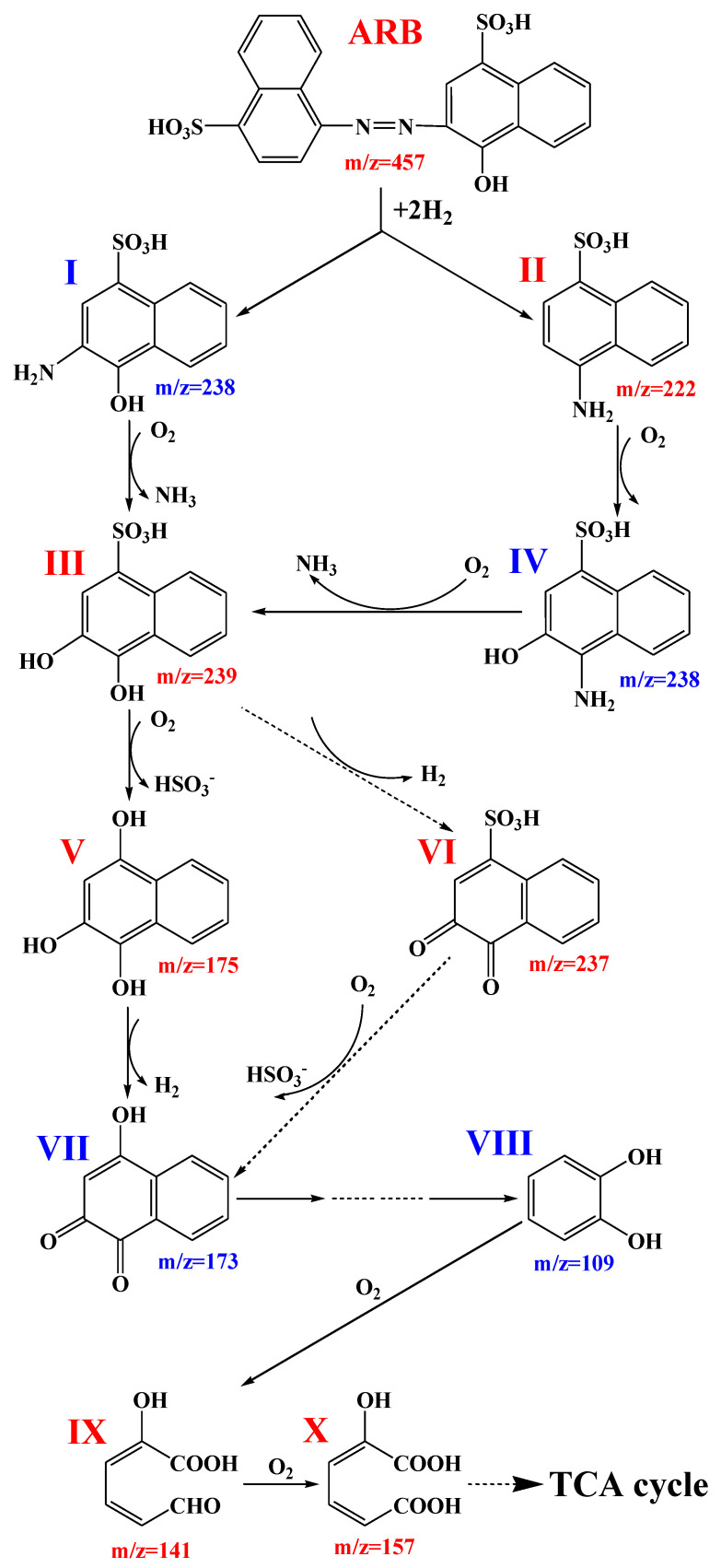
Possible decolorization pathways of ARB by *M. guilliermondii* A4. I, 3-amino-4-hydroxynaphthalene-1-sulfonic acid; II, 4-aminonaphthalene-1-sulfonic acid; III, 3,4-dihydroxynaphthalene-1-sulfonic acid; IV, 4-amino-3-hydroxynaphthalene-1-sulfonic acid; V, naphthalene-1,2,4-triol; VI, 3,4-dioxo-3,4-dihydronaphthalene-1-sulfonic acid; VII, 4-hydroxynaphthalene-1,2-dione; VIII, pyrocatechol; IX, (2*E*,4*Z*)-2-hydroxy-6-oxohexa-2,4-dienoic acid; X, (2*E*,4*Z*)-2-hydroxyhexa-2,4-dienedioic acid.

**Figure 6 jof-09-00851-f006:**
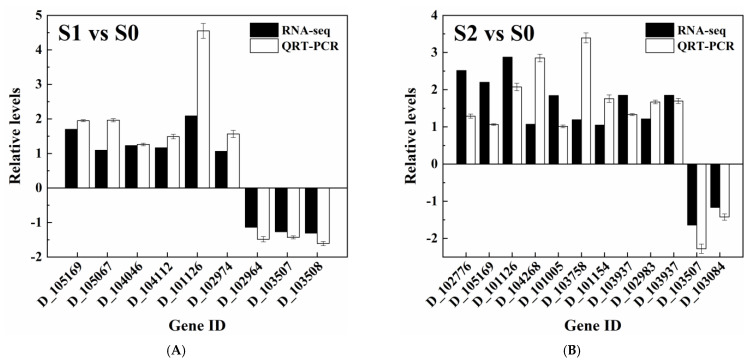
QRT-PCR validation of the relative expression abundance of selected DEGs in (**A**) “S1 vs. S0” and (**B**) “S2 vs. S0” groups.

**Figure 7 jof-09-00851-f007:**
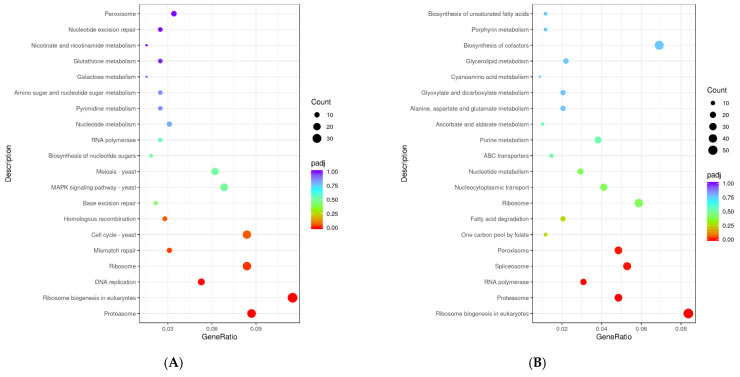
Enrichment of KEGG pathways in the comparisons of (**A**) S1 and S0, and (**B**) S2 and S0.

**Table 1 jof-09-00851-t001:** DEGs related to halotolerance and key enzymes of *M. guilliermondii* A4 in “S1 vs. S0” and “S2 vs. S0” comparisons.

Description	Gene ID	Annotation (Protein ID)	log_2_ FC **	
			S1 vs. S0	S2 vs. S0
Cell wall regulation	D_102776	Essential for maintenance of the cell wall protein 1 (P42842)	——	2.514 *
	D_105169	1,3-beta-glucan synthase component GSC2 (P40989)	1.697 *	2.201 *
	D_105067	1,3-beta-glucan synthase component FKS3 (Q04952)	1.092 *	——
	D_104046	Endo-1,3(4)-beta-glucanase 1 (Q5AIR7)	1.228 *	——
	D_104112	Cell wall protein ECM33 (C7GQJ1)	1.165 *	——
	D_101126	Cell surface GPI-anchored protein ECM33 (Q5AGC4)	2.088 *	2.877 *
	D_101744	Chitin synthase 3 (P30573)	1.514	2.918
	D_100586	Chitin synthase 2 (P30572)	1.160	——
	D_100803	Chitin synthase 1 (P23316)	——	1.638
	D_103766	Chitin synthase 2 (P30572)	——	1.405
Potassium transporter (high-salt-in strategy)	D_102974	High-affinity potassium transporter (P50505)	1.063 *	——
	D_104268	Low-affinity K^+^ transporter 1 (P47114)	——	1.071 *
Compatible solutes (organic-solute-in strategy)	D_103937	NADP-specific glutamate dehydrogenase (P29507)	——	1.850 *
	D_102983	Glycerol-3-phosphate dehydrogenase, mitochondrial (O14400)	——	1.213 *
	D_102964	Glycerol-3-phosphate dehydrogenase [NAD^+^] (Q7ZA43)	−1.134 *	——
	D_103937	NADP-specific glutamate dehydrogenase (P29507)	——	1.850 *
Other possible halotolerant mechanisms	D_101005	Calcium-transporting ATPase 1 (P13586)	——	1.841 *
	D_103758	Calcium-transporting ATPase 2 (P38929)	——	1.193 *
	D_101154	Casein kinase II subunit alpha (P21868)	——	1.050 *
Possible key enzymes	D_103507	Probable NADPH dehydrogenase (NADH:flavin oxidoreductase) (P43084)	−1.268 *	−1.637 *
	D_103508	Probable NADPH dehydrogenase (NADH:flavin oxidoreductase) (P43084)	−1.308 *	——
	D_103084	YEAST manganese transporter SMF1 (P38925)	−0.178	−1.165 *
	D_100122	Laccase PFICI_06862 (W3X732)	0.402	−0.283
	D_104858	Cytochrome c peroxidase (Q6BKY9)	0.678	−0.078

* Relative expression abundance was validated through QRT-PCR (as shown in Figure 6); ** the corresponding FDR was less than 0.05.

## Data Availability

All data generated or analyzed during this study are included in this published article and its Appendix A. Partial 26S rDNA and ITS sequences of *M. guilliermondii* A4 were deposited to the GenBank database with the accession numbers of ON102041 and ON089018, respectively. All raw transcriptome data were deposited in the National Center for Biotechnology Information (NCBI) with the BioProject ID of PRJNA961313.

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
