# Peer review of "A Potentially Practicable Halotolerant Yeast Meyerozyma guilliermondii A4 for Decolorizing and Detoxifying Azo Dyes and Its Possible Halotolerance Mechanisms"

_jof, 2023, doi:10.3390/jof9080851_

Round 1
Reviewer 1 Report
Abstract: The information provided gave a general background of the study, but it could be improved to provide a more concise and coherent overview.
Introduction:
Overall, this section comprehensively establishes the study's significance by highlighting the issue of textile dyeing wastewater, focusing on azo dyes and their environmental impact. It discusses the advantages of biological methods, particularly microorganisms, for treatment. The limitations of using bacteria and fungi lead to exploring yeasts as potential alternatives, especially halotolerant ones due to high salinity in dyeing wastewater. The study's motivation is to address the scarcity of effective halotolerant yeasts for pollutant treatment, necessitating further research. Overall, the background provides a solid foundation for understanding the study's context and significance.
Minor edits to improve the write-up are suggested as follows:
Line 35: are usually difficult to treat
Line 39: possessing > 70%
Line 40: Due to the failure to bind to fibers
Line 41: about 10% of dyestuff
Line 42: are harmful to aquatic life
Line 43: azo dyes' degradation
Line 44: and may also be mutagenic and carcinogenic
Line 45: remove residual azo dyes from wastewaters and detoxify them before releasing them into the environment
Line 52: easy to obtain
Line: 55 However, the toxic decolorization intermediates of azo dyes (e.g. aromatic amines) can inhibit the vast majority of bacteria, thus leading to the failure of purification and even an increase in biotoxicity after treatment
Line 65: As a class of single-cell fungi
Line 94: they do not survive depending on certain salinity
Material and methods:
This section requires clarification and the addition of missing information to explain the proper methodology that was carried out in this project.
Section 2.1 just listed all the dyes used without justifying the selection. Why were these seven dyes selected? I suggest listing the furnisher in brackets after every dye so it is more concise.
Line 134: The selective culturing medium contains – Please include the name of the media used and also the furnisher for the agar. What was the incubation setting? The isolation methods should be mentioned briefly before citing the reference.
Was the strain isolated from field samples? Please provide some details on the site where the strains were originally retrieved. Section 2.2 requires some additional detail. At present, it is unclear to the reader about the origin of these isolates.
Line 154: Concentrations of the azo dyes and yeast cell suspension
Line 164: Remove "be" -which were probably responsible
Line 169: were shown in a previous report
Line 174: spectrophotometry ; High-performance liquid chromatography – mass spectrometry (HPLC-MS)- please include furnisher for both
Line 200: p-value
Results and discussion
The results were well elaborated but some sections have extremely long paragraphs. I suggest separating each point into smaller paragraphs so it is easier to follow. The discussion could be elaborated. Comments are as listed below:
Line 206-226: This section can be separated into paragraphs so the reader can follow the transition from on point to another. Morphological identification (para1 ), genetic evidence (para 2) and so on…
Line 206: Please add a comma after A4
Line 209: edge was misspelled
Line 212: Transcribed Space should be Transcribed Spacer
Line 220: ‘Five’ of them including ARB, 3R, AOII – no mention of these previously.
Line 279: ‘nigh yeast strains’- please check your spelling. Ditto line 285
Line 302: Please elaborate on this - ‘competition between the generation and further degradation of the toxic dye decolorization intermediates in different yeasts’.
Line 390: This paragraph is very long and can be separated into a few paragraphs so it is clearer
Line 437: was this conclusion derived from information that were available from previous work? If yes, please include your citation
Line 461: Though no DEG encoding LiP was found in any of the three groups of samples..
Line 465: ‘The possible reason might be the incomplete annotation of the reference genome.’ -What was the reason behind this? Please elaborate
The quality of English is generally acceptable but can be significantly improved
Reviewer 2 Report
The purpose of the manuscript is to identify a method of biological treatment for water that is polluted with azo dyes. For this, the authors isolate a marine ýeast and characterize it based on its ability to detoxify azo dyes. Besides the ability to act on dyes the authors also investigate the yeast’s property of halotolerance, as conditions in wastewater that contains dyes are hipersaline. The authors have previously published several other papers similar to the current one, where yeast were characterized based on their ability to degrade azo dyes, some of them were also halotolerant. Based on this the authors have the protocols and experimental design well set up and the experiments are well controlled and executed. However, the fact that they have done the same many times over makes this paper unoriginal. It is not clear from the manuscript why they decided to look for more yeast that can detoxify azo dyes besides the fact that they can and have the protocols set up. The authors need to bring arguments as to why this new yeast is better or different than the ones previously described and what this study brings new to the previous ones they have published.
Please provide the appropriate reference for the methods. You mention "method described by Li et al. [20]." however reference 20 is "20. Kavynifard, A.; Ebrahimipour, G.; Ghasempour, A. Optimization of crude oil degradation by dietzia cinnamea KA1, capable of 603 biosurfactant production. J. Basic Microbiol. 2016, 56, 566–575."
There is a Li et al reference, number 24, but that one is a review article so I am not sure that is the correct one for the references either. Please make sure you number the references correctly and you have the appropriate references. The rest of the references need to be checked to make sure there are no other mistakes.
Overall the manuscript is well thought out, the experiments are sound and well controlled but it does not bring new information to the field, or just marginally new. A major improvement that can be made to the article is to make clear what makes this article different from the rest and what new and relevant information it brings.
The article would also benefit from English editing.
Please have the English checked by a native speaker as it needs improving.
